# Effects of E’Jiao on Skeletal Mineralisation, Osteocyte and WNT Signalling Inhibitors in Ovariectomised Rats

**DOI:** 10.3390/life13020570

**Published:** 2023-02-17

**Authors:** Kok-Yong Chin, Ben Nett Ng, Muhd Khairik Imran Rostam, Nur Farah Dhaniyah Muhammad Fadzil, Vaishnavi Raman, Farzana Mohamed Yunus, Wun Fui Mark-Lee, Yan Yi Chong, Jing Qian, Yan Zhang, Haibin Qu, Syed Alhafiz Syed Hashim, Sophia Ogechi Ekeuku

**Affiliations:** 1Department of Pharmacology, Faculty of Medicine, Universiti Kebangsaan Malaysia, Jalan Yaacob Latif, Bandar Tun Razak, Cheras 56000, Malaysia; 2Department of Chemistry, Faculty of Science, University Teknologi Malaysia (UTM), Johor Bahru 81310, Malaysia; 3Research Center for Quantum Engineering Design, Department of Physics, Faculty of Science and Technology, Universitas Airlangga, Jl. Mulyorejo, Surabaya 60115, Indonesia; 4School of Pre-University Studies, Taylor’s College, Subang Jaya 47500, Malaysia; 5Pharmaceutical Informatics Institute, College of Pharmaceutical Sciences, Zhejiang University, Hangzhou 310030, China

**Keywords:** Colla corii asini, Dickkopf-1, sclerostin, parathyroid hormone, osteoporosis

## Abstract

E’Jiao is a traditional Chinese medicine derived from donkey skin. E’Jiao is reported to suppress elevated bone remodelling in ovariectomised rats but its mechanism of action is not known. To bridge this research gap, the current study aims to investigate the effects of E’Jiao on skeletal mineralisation, osteocyte and WNT signalling inhibitors in ovariectomised rats. Female Sprague–Dawley rats (3 months old) were ovariectomised and supplemented with E’Jiao at 0.26 g/kg, 0.53 g/kg and 1.06 g/kg, or 1% calcium carbonate (*w*/*v*) in drinking water. The rats were euthanised after two months of supplementation and their bones were collected for Fourier-transform infrared spectroscopy, histomorphometry and protein analysis. Neither ovariectomy nor treatment affected the skeletal mineral/matrix ratio, osteocyte number, empty lacunar number, and Dickkopf-1 and sclerostin protein levels (*p* > 0.05). Rats treated with calcium carbonate had a higher Dickkopf-1 level than baseline (*p* = 0.002) and E’Jiao at 0.53 g/kg (*p* = 0.002). In conclusion, E’Jiao has no significant effect on skeletal mineralisation, osteocyte and WNT signalling inhibitors in ovariectomised rats. The skeletal effect of E’Jiao might not be mediated through osteocytes.

## 1. Introduction

Osteoporosis is a skeletal disease marked by reduced bone mineral density and strength, which predisposes patients to low-impact fragility fractures [1]. Osteoporosis is more common among postmenopausal women and the elderly [2]. A recent meta-analysis revealed that 23.1% of the female population [95% (confidence interval) CI 19.8–26.9%] and 11.7% of the male population (95% CI 9.6–14.1) worldwide have osteoporosis [3]. This observation is most probably due to oestrogen deficiency secondary to menopause in women [4]. The ovarian function ceases during menopause, resulting in decreased oestrogen secretion and a concurrent rise in follicle-stimulating hormone levels, which produce a rapid bone loss phase in postmenopausal osteoporosis [5].

Osteoporosis occurs as a result of the imbalance bone remodelling process, favouring bone loss [6]. Bone remodelling is important in maintaining the integrity and functions of bone tissues, whereby osteoblasts, osteoclasts and osteocytes work sequentially in governing this biological process throughout life [7]. Osteocytes are the master regulator of bone remodelling and the most populous cells in the bone [8,9]. They could sense mechanical loading and mediate bone formation by osteoblasts and bone resorption by osteoclasts [10]. The osteocytes serve as a significant source of receptor activator of nuclear factor kappa-B ligand to support osteoclastogenesis, and in secreting the bone formation inhibitor sclerostin (SOST) and Dickkopf-1 (DKK-1) [11]. Oestrogen suppresses bone remodelling activation partially through osteocytes. Oestrogen also restricts bone resorption by acting on osteoclasts directly or modulating osteoclast formation and activity mediated by osteoblast/osteocyte- and T-cell [12].

The canonical WNT pathway is one of the signalling pathways affecting bone remodelling. Its activation provides skeletal anabolic effects [13]. β-catenin is the critical signalling molecule in the canonical WNT signalling pathway. Without WNT stimulation, phosphorylation of cytoplasmic β-catenin occurs through adenomatous polyposis coli/glycogen synthase kinase-3/axin complex, causing it to be degraded by the proteasome system [14]. WNT signalling pathway are inhibited by SOST and DKK-1 endogenously [15,16]. DKK-1 binds to the low-density lipoprotein receptor-related protein (LRP) 5/6 receptor and forms a complex with Kremen at the cellular surface, causing internalisation of LRP5/6 receptor and inhibition of the canonical WNT pathway. SOST binds to LRP 5/6 and LRP4, which enhances its suppressive effects on the canonical WNT signalling pathway [14]. The inactivation of the WNT signalling pathway causes an increase in bone resorption and a decrease in bone formation leading to reduced bone mass [17] (Figure 1).

The current pharmacological agents for osteoporosis target various signalling pathways involved in bone remodelling to achieve their therapeutic effects. They can be classified into antiresorptive agents and anabolic agents. Examples of antiresorptive agents are bisphosphonates, denosumab and selective oestrogen receptor modulators, while teriparatide and romosozumab are examples of anabolic agents [18]. These agents can increase bone mineral density and reduce fracture risk, but they come with undesirable effects. Calcium and/or vitamin D are the most common osteoporosis preventive agents [19]. Hence, there is a demand for newer pharmacological agents to maintain and restore bone health. The search is expanded to natural products and traditional medicine.

E’Jiao, also known as Colla corii asini, is a traditional Chinese medicine derived from donkey-hide gelatine [20,21,22]. Modern pharmacology studies demonstrated that E’Jiao could significantly increase haematopoietic stem cells, burst-forming unit-erythroid and colony-forming unit granulocyte-macrophages in the bone marrow. E’Jiao could also promote the differentiation of haematopoietic progenitor cells to all lineages, which might accelerate the recovery of haematopoietic function in myelosuppressive conditions [23,24]. Of note, E’Jiao is suggested to exert its biological effects through WNT signalling and its regulatory proteins SOST and DKK-1 [24]. In vitro studies have shown that E’Jiao can promote osteoblast differentiation, characterised by increased expression of alkaline phosphatase and bone morphogenetic protein-2 expressions. These effects were mediated by the phosphoinositide 3-kinase/Akt/nuclear factor kappa-light-chain-enhancer of activated B-cell signalling pathway [25]. A study on ovariectomised (OVX) rats found that E’Jiao could suppress high bone remodelling phenotype due to oestrogen deficiency [26]. However, the mechanism of suppression is not known.

Given its potential effects on SOST and DKK-1, it is postulated that E’Jiao might affect osteocytes that secrete these proteins in regulating bone health. Therefore, the current study aimed to investigate the effects of E’Jiao on the osteocyte population and skeletal DKK-1 and sclerostin SOST protein levels in OVX rats. The effects of E’Jiao on the mineralisation of skeletal tissue were also studied through Fourier-transform infrared spectroscopy (FTIR), which is a technique used to reveal bone material properties [27]. We hope this study could further the understanding of the mechanism of E’Jiao as a potential anti-osteoporotic agent among postmenopausal women.

## 2. Materials and Methods

### 2.1. Treatment Agent

The E’Jiao utilised in this study was a gift from Shandong Dong-E-E-Jiao Co., Ltd. (Dong-e, China), with an indicated human dose of 6 g/day. The chemical characterisation of this formulation has been reported [28]. With an assumption of body weight of 60 kg for an adult human, it could be converted to 0.1 g/kg body weight/day. Based on the body surface ratio, the dose for rats was estimated to be 0.53 g/kg body weight/day [29]. Half and two times of this dose were used to establish a dose–response relationship. A previous study reported that even at half the recommended dose, E’Jiao could prevent high bone remodelling in ovariectomised rats [26]. The E’Jiao was dissolved in distilled water for oral gavage. Calcium carbonate (May & Baker Ltd., Dagenham, England) was used as the positive control in this study. The dose used (1% in drinking water) has been shown to prevent osteoporosis in multiple osteoporosis models [30,31].

### 2.2. Animals and Treatment

Forty-two female Sprague–Dawley rats (three-month-old) sourced from the Laboratory Animal Research Unit, Universiti Kebangsaan Malaysia (Kuala Lumpur, Malaysia) were housed in the animal holding facility at the Department of Pharmacology, Universiti Kebangsaan Malaysia under standard conditions (27 ± 1 °C, ambient humidity, 12:12 light:dark cycle). They were given ad libitum access to food (702P standard rat chow, Goldcoin, Klang, Malaysia) and tap water. After one week of acclimatisation, they were divided randomly into 7 groups. The baseline group was sacrificed before any treatment. Bilateral ovariectomy was performed on all groups except the sham to induce oestrogen deficiency. Laparotomy was performed on the sham group, but ovaries were not removed. The supplementation was initiated 7 days after surgery to allow the rats to recover from the surgery. The sham and OVX control received tap water. The E’Jiao-L, -M and -H groups received E’Jiao solution at the dose of 0.26 g/kg, 0.53 g/kg and 1.06 g/kg, respectively, through oral gavage daily. The calcium control group received 1% calcium carbonate dissolved in drinking water. After two months of treatment, the rats were euthanised, and their bones were harvested for analysis. A similar animal treatment protocol has been reported elsewhere [26].

### 2.3. Measurement of Osteocyte and Empty Lacunar Number

The right femur was sawed into equal pieces and preserved in 70% formaldehyde. The bone samples were decalcified for two months at room temperature using a 20% ethylenediaminetetraacetic acid solution (Sigma-Aldrich, St. Louis, MO, USA). Following decalcification, samples were sectioned using a microtome (Leica RM2235, Nussloch, German) with a Leica 818 high-profile microtome blade and embedded in paraffin (Leica Biosystems Richmond Inc., Richmond, IL, USA). Haematoxylin-eosin (H&E) (Abcam, Cambridge, UK) staining was performed on these sections to see the osteocytes and lacunae. Using the Weibel grid approach, the osteocyte number and empty lacunar number were calculated [32].

### 2.4. FTIR Spectral Acquisition and Band Area Analysis

Another sagittal half of the femur was not calcified. It was embedded in polymerised resin (Osteo-Bed bone embedding kit, Sigma-Aldrich, St. Louis, MO, USA) for preservation, and ground to powder for FTIR analysis. The chemical composition of the compacted powder was measured on Agilent Cary 630 spectrophotometer equipped with a 5-bounce zinc selenide attenuated total reflectance (ZnSe ATR) sampling accessory (Agilent Technologies, Santa Clara, CA, USA). The ZnSe ATR sample interface was cleaned with deionised water and acetone before spectrum collection. The resin powder without bone sample served as blank during the measurement (64 scans). The absorption range between 4000 and 650 cm^−1^ with 64 scans was measured at room temperature with a resolution of 4 cm^−1^. The area under the region 890–1212 cm^−1^ for phosphate v1v3 and 1680–1780 cm^−1^ for amide I was quantified as mineral /matrix ratio to reflect the mineral content in relationship to collagen content of the bone [33]. The spectral band area calculation was performed with SpectraGryph, version 1.2.16.1 [34].

### 2.5. Measurement of WNT Inhibitors

The left tibia was stripped of soft tissues and then homogenised in liquid nitrogen using a mortar and pestle. Using RIPA lysis buffer, proteins from the bone samples were extracted (Elabscience Biotechnology, Houston, TX, USA). The homogenates were centrifuged at 8000 g for 10 min, and the supernatant was used to assay the total protein levels using the Bradford technique (Quick StartTM Bradford Protein Assay, Bio-Rad Laboratories Inc., Hercules, CA, USA). Following the manufacturer’s instructions, Milliplex^®^ Map Kits (EMD Millipore Corporation, Billerica, MA, USA) were used to measure the protein levels of DKK-1 and SOST. Luminex 100 (Luminex Corporation, Austin, TX, USA) was used to measure the signals. The concentration of total proteins in each sample was used to normalise the levels of these proteins. The methods of measurement are based on the study of Wong et al. (2019) [35].

### 2.6. Ethical Consideration

Universiti Kebangsaan Malaysia Animal Ethics Committee (approval code: FAR/2022/KOK YONG/23-MAC/1235-MAC-2022-NOV.-2022-NAR-CAT2) reviewed and approved the protocol of the current study. The procedures were performed in accordance with the Malaysian Animal Welfare Act (2015).

### 2.7. Statistical Analysis

Statistical Package for Social Sciences version 23.0 was used to conduct the statistical analysis (IBM, Armonk, NY, USA). The Shapiro–Wilk test was used to determine whether the data were normal. One-way analysis of variance was used to compare normally distributed data (osteocyte number, empty lacunar number, and SOST levels) before pairwise post hoc comparison. The Kruskal–Wallis test, followed by the Mann–Whitney U-test with Bonferroni adjustment, was used to analyse skewed data (DKK-1 levels). Statistical significance was defined as a *p*-value < 0.05.

## 3. Results

### 3.1. Osteocyte and Empty Lacunar Number

Bone histomorphometry analysis revealed a significantly higher trabecular osteocyte number in the baseline group compared to the sham group (*p* = 0.04). However, no significant difference in osteocyte and empty lacunar number of cortical and trabecular bone was observed between OVX and sham (*p* > 0.05) group. Similarly, all treatments did not alter the osteocyte and empty lacunar number of both bone compartments (*p* > 0.50 vs. OVX group) (Figure 2 and Figure 3).

### 3.2. FTIR Analysis

The representative FTIR spectrum of each group is presented in Figure 4A. FTIR analysis revealed that neither ovariectomy nor any of the treatment agents affected the skeletal mineral/matrix ratio of the rats (*p* > 0.05) (Figure 4B).

### 3.3. Bone Biochemical Markers

Ovariectomy did not affect the protein level of DKK-1 in the bone (*p* > 0.05 vs. sham group). However, the rats treated with calcium carbonate had a significantly higher DKK-1 level than the baseline group (*p* = 0.002) and the rats treated with E’Jiao at the moderate dose (*p* = 0.002). The protein level of SOST was not significantly different between the baseline and the sham group, and between the OVX and the sham group (*p* > 0.05). All treatments also did not affect the protein level of SOST in the bone (*p* > 0.05 vs. OVX group) (Figure 5A,B).

## 4. Discussion

The current study found that ovariectomy did not alter osteocyte and empty lacunar number of trabecular and cortical bone in female rats. Besides, it also did not affect skeletal mineral/matrix ratio, DKK-1 and SOST levels in these rats. E’Jiao at all tested doses and calcium carbonate also did not affect all indices tested in this study.

### 4.1. The Effects of Ovariectomy and E’Jiao on Osteocyte Parameters

The effects of ovariectomy on osteocyte number and WNT signalling inhibitors observed by this study were different from previous literature. Emerton et al. (2009) observed increased osteocyte apoptosis in OVX rats, wherein the number of apoptotic osteocytes was four- to seven-fold higher than sham group 3 days after ovariectomy [36]. Mohamad et al. (2021) reported that the osteocyte number in 12-month-old OVX rats was significantly lower than in the sham group 4 months after ovariectomy [32]. In the same study, ovariectomy increased SOST protein level but not DKK-1 [32]. The possible reasons for this discrepancy include the difference in the age of our rats compared to previous models. The rats in our study were 5 months old, while in the study of Mohamad et al. (2021) their rats were 12 months old at the end of the study [32]. We postulated that younger rats are less susceptible to the effect of oestrogen deficiency induced by ovariectomy than older rats. Other than that, the duration of the oestrogen deficiency induced by ovariectomy is only 2 months in our study, which may not be sufficient to cause osteocytic changes in the rats.

The current study found no effects of E’Jiao on osteocyte number and WNT signalling inhibitors. The effects of E Jiao on osteocyte number have not been reported in the literature, thus there was no study that we could compare to. On the other hand, Zhang et al. (2019) reported that E’Jiao might regulate WNT signalling by decreasing the expression of SOST and DKK-1 in mice treated with 5-fluorouracil [24]. The discrepancy between this study and their findings might be due to a difference in the model used, but this awaits further validation.

In the current study, the DKK-1 protein level increased in the calcium carbonate-supplemented group. This might be due to the increase in active osteocytes number. This increase was not reflected through histomorphometric methods maybe because only a few bone histology sections were counted for each rat. On the other hand, the whole tibia bone was homogenised for DKK-1 estimation, and it could reflect the osteocyte number or activity better. A previous study reported that acute exposure to parathyroid hormone (PTH) reduced DKK-1 mRNA expression in cultured bones [37]. In extrapolation of this finding, rats supplemented with calcium might have low circulating PTH levels due to the negative physiological feedback loop, thus DDK-1 expression was elevated in the bone. However, we did not determine the serum PTH level in this study.

### 4.2. The Effects of Ovariectomy and E’Jiao on Mineral/Matrix Ratio

The mineral/matrix ratio revealed by FTIR analysis is biologically relevant because it considers both the mineral and organic components of the bone. It could indicate whether a bone is hypomineralised due to osteoporosis [27]. A previous study reported a reduced mineral/matrix ratio in ovariectomised rats (8-week-old) treated with cortisol (10 mg/kg), which suffered from severe bone loss [38]. This reduction was not observed in our case, probably because the bone loss in the rats with ovariectomy per se was milder compared to the aforementioned study. In parallel, Ekeuku et al. (2022) showed that oestrogen deficiency and E’Jiao treatment for two months did not affect the bone mineral density and content of the rats [26].

### 4.3. Limitations of the Study

The current study is limited by the young age of the rats. Castrated sexually matured female rats have been used frequently as a model of osteoporosis but they might be reflective of a stunted bone growth model rather than an osteoporosis model [39,40]. The use of older rats might represent a better model. The period of supplementation for E’Jiao could be prolonged in the future to achieve better effects. Nevertheless, the findings of this study might also hint that the skeletal effects of E’Jiao might not be osteocyte or WNT-dependent. Future studies could explore other mechanisms to explain the skeletal beneficial effects of E’Jiao.

## 5. Conclusions

E’Jiao does not have significant effects on osteocyte number and WNT signalling inhibitors, i.e., SOST and DKK-1, secreted by osteocytes in OVX rats. The skeletal effect of E’Jiao might not be mediated through osteocytes or WNT signalling pathway. Future studies should examine other mechanisms to explain the anti-osteoporosis effects of E’Jiao.

## Figures and Tables

**Figure 1 life-13-00570-f001:**
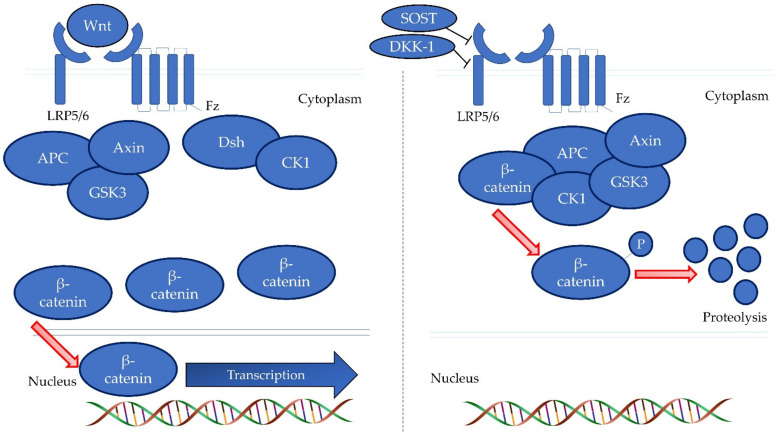
The canonical Wnt signalling pathway. The binding of SOST and DKK-1 with LRP receptors prevents proteolysis of β-catenin, thus preventing nuclear translocation of β-catenin and transcription of genes coding for bone formation. Abbreviation: APC, adenomatous polyposis coli; CK1, Casein kinase 1; Dsh, dishevelled proteins; DKK-1, Dickkopf-1; Fz, frizzled proteins; GSK3, glycogen synthase kinase-3; LRP5/6, low-density lipoprotein receptor-related protein; SOST, sclerostin.

**Figure 2 life-13-00570-f002:**
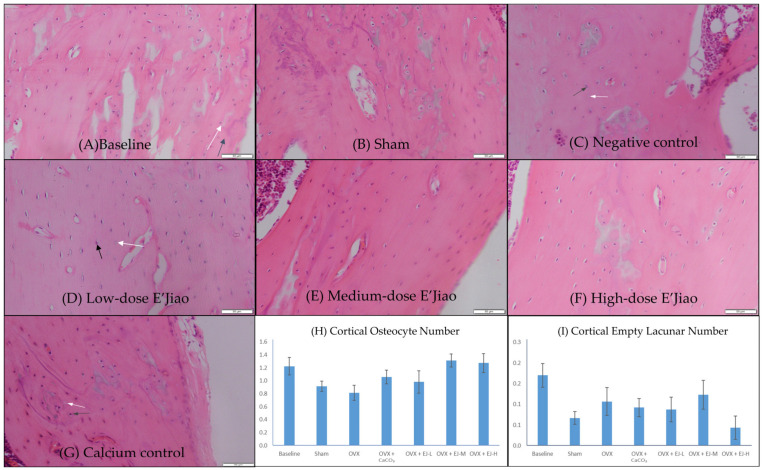
Haematoxylin and eosin-stained femur sections histological slides of cortical bone (**A**–**G**) for quantification of osteocyte number (**H**) and empty lacunar number (**I**). The values are expressed as mean ± standard error mean (*n* = 6 rats in each group). Abbreviations: OVX, Ovariectomy; CaCO_3_, calcium carbonate; EJ-L, E’Jiao-low dose; EJ-M, E’Jiao-medium dose; EJ-H, E’Jiao-high dose. Notes: The white arrow shows the empty lacunae, while the black arrow shows the osteocytes.

**Figure 3 life-13-00570-f003:**
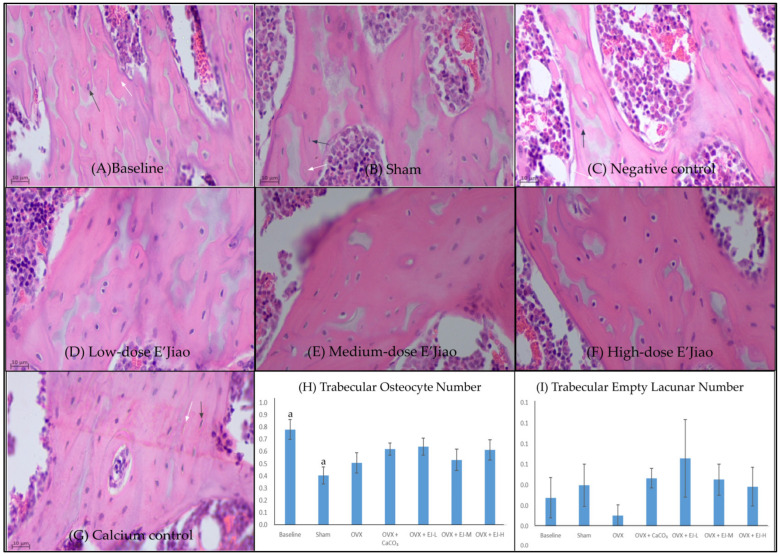
Haematoxylin and eosin-stained femur sections histological slides of trabecular bone (**A**–**G**) for quantification of osteocyte number (**H**) and empty lacunar number (**I**). The values are expressed as mean ± standard error mean (*n* = 6 rats in each group). Abbreviations: OVX, Ovariectomy; CaCO_3_, calcium carbonate; EJ-L, E’Jiao-low dose; EJ-M, E’Jiao-medium dose; EJ-H, E’Jiao-high dose. Notes: The white arrow shows the empty lacunae, while the black arrow shows the osteocytes. The groups sharing the same alphabet are significantly different from each other.

**Figure 4 life-13-00570-f004:**
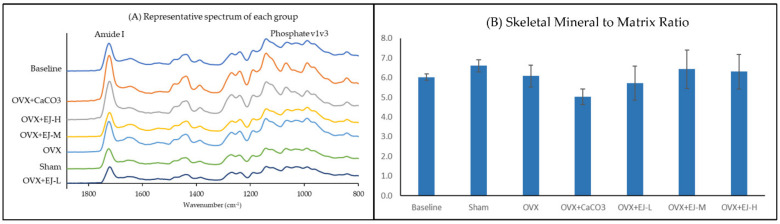
FTIR analysis of mineral/matrix ratio of bone sample from each group. The representative spectrum of each group is shown in (**A**), while the quantitative comparison of the mineral/matrix ratio among the group is shown in (**B**). The values are expressed as mean ± standard error mean (*n* = 6 rats in each group) in (**B**). Abbreviations: OVX, Ovariectomy; CaCO_3_, calcium carbonate; EJ-L, E’Jiao-low dose; EJ-M, E’ Jiao-medium dose; EJ-H, E’ Jiao-high dose.

**Figure 5 life-13-00570-f005:**
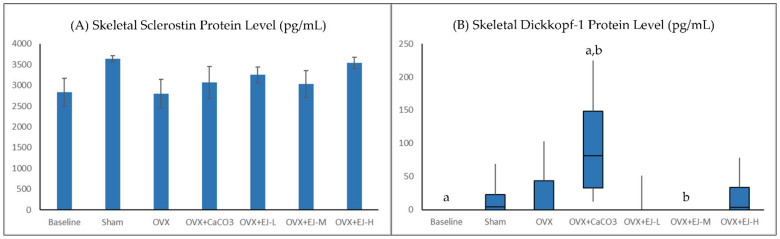
Skeletal sclerostin (**A**) and Dickkopf-1 protein levels (**B**) of each group. Skeletal sclerostin protein levels are expressed as mean ± standard error mean (*n* = 6 rats in each group). Skeletal Dickkopf-1 protein levels are presented in box plots because the data are skewed and analysed using the Kruskal–Wallis test. Abbreviations: OVX, Ovariectomy; CaCO_3_, calcium carbonate; EJ-L, E’Jiao-low dose; EJ-M, E’ Jiao-medium dose; EJ-H, E’ Jiao-high dose. The groups sharing the same alphabet are significantly different from each other.

## Data Availability

The data are available at reasonable request from the corresponding author.

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
