# Peer review of "Effects of E’Jiao on Skeletal Mineralisation, Osteocyte and WNT Signalling Inhibitors in Ovariectomised Rats"

_life, 2023, doi:10.3390/life13020570_

Round 1

Reviewer 1 Report

 E’Jiao is a famous chinese supplement. Clinical result in promoting musculoskeletal health is proven with a good track record. It is a pity author can not prove it with a pathway in your research. However, the overall study design is still solid. The article should meet the publication requirement after minor revision and cite more article from life journal...

Author Response

Dear reviewer,

Thank you for reviewing our manuscript. We appreciate the constructive comments given and have responded to each of them in the attached response sheet.

Reviewer 2 Report

1.In page 1, part" abstract"

Why the authors have not introduced "E’Jiao"?

2. In page 1, part" abstract"

Why the authors have not spoken about the importance of their present work in this part?

3. page 1, part" abstract"

Why the authors have mentioned detailed results? It is not necessary to remark every detailed part of your results in this part.

4. Page 1, part" abstract"

The authors have written about their material and methods with detail in this part. Please note that the part "abstract" should introduce the total concept about the manuscript. 

5.In page 1, part" abstract"

Why the authors have not told about the conclusion of the manuscript briefly?

6.In page 1, line 42, part" key words"

Why the part "key words" has not contained " E’Jiao" and "WNT pathway"?

7. In page 2, line 56

Please reform multiple references in this part.

 7.Page 1, line 45-56 the authors have written about "Osteoporosis" but in the next paragraph which is located in page 2, line 57-67, they have written about" Bone remodelling" without any link between these two paragraphes. Please keep consistency of the text of the manuscript.

8. In page 2 

Please create a link between two paragraphes that are located in line 57-67 

and line 68-79 or rewrite them in order to keep continuity of the text.

9. In page 2, line 68-79

Why the authors have not drawn a simple figure to show all of the mechanisms that have been mentioned in this part of the manuscript? It would help other readers to understand this part better.

10. In page 2, line 80-86

This paragraph has not any logical link with its prior paragraph. Please consider this note and reform this part based on mentioned note.

11. In page 2, line 92

Why authors have inserted multiple references in this part?

12. In page 3, line 101

Please reform the middle-sentence reference

13. In page 3, line 112-113

What is the purpose of the sentence"The chemical characterisation of this formulation has been reported [25]. " in this part of the manuscript?

14. In page 3, line 114

Please reform the middle-sentence reference in this part.

15. In page 3, line 116-118

Why the authors have mentioned this sentence" A previous study reported that even at half the recommended dose, E’Jiao could prevent high bone remodelling in ovariectomised rats [23]. " in the part "material and methods" ??

16. In page 3, line 118

Please add necessary information about "Calcium carbonate " that used in the manuscript (where it has been purchased, the name of company and etc.)

17. Page 3, line 119-120

Why the authors have written the sentence"The dose used (1% in drinking water) has been shown to prevent osteoporosis in multiple osteoporosis models [27,28]. " in this part?

18. In page 3, line 121

Please tell me that are you performed all of mentioned processes in this part based on previous scientific studies of other reseaechers or yourselves? 

If you performed based on prior scientific works, please add its name with proper reference.

19. Page 3, line 146

Why the authors have inserted two references here?

20. In page 4, line 158 and 159

Please tell me the reason of two citation of [31] and [32] in this part.

21. Please tell the scientific reference that you performed the part"2.5 Measurement of WNT inhibitors" based on it.

22. In page 6, line 216

Is it necessary to mention the sentence "The comparison is performed using one-way analysis of variance." In this part?

23. In page 6, line 228-234

Please omit the sentence" and analysed using the one-way analysis of variance" in line 229-230

Besides, some abbreviations are repetitive 

(you have mentioned them in the title of prior figures) please reform them.

24. About the part "Discussion" in page 7

Please categorize your findings from the most important to the least important one in the form of subheadings for this part. Then discuss about each one of them. 

Note that these subheadings should contain some data about the limitaions of the present panuscript and the comparisons of the present study with prior related researches.

25. Please check reference list (specifically their title?

Author Response

(The authors gave the same response as above.)

Round 2

Reviewer 2 Report

Thanks to the respected authors. The article can be published under the current conditions and I have no other suggestions or comments.

Author Response

Thank you for the comment. There is no specific comment to be addressed.